# Gut-to-Brain α-Synuclein Transmission in Parkinson’s Disease: Evidence for Prion-like Mechanisms

**DOI:** 10.3390/ijms24087205

**Published:** 2023-04-13

**Authors:** Merry Chen, Danielle E. Mor

**Affiliations:** Department of Neuroscience and Regenerative Medicine, Medical College of Georgia, Augusta University, Augusta, GA 30912, USA

**Keywords:** alpha-synuclein, Parkinson’s disease, enteric nervous system, prion-like

## Abstract

Parkinson’s disease (PD) is a multifactorial disorder involving both motor and non-motor symptoms caused by the progressive death of distinct neuronal populations, including dopaminergic neurons in the substantia nigra. The deposition of aggregated α-synuclein protein into Lewy body inclusions is a hallmark of the disorder, and α-synuclein pathology has been found in the enteric nervous system (ENS) of PD patients up to two decades prior to diagnosis. In combination with the high occurrence of gastrointestinal dysfunction in early stages of PD, current evidence strongly suggests that some forms of PD may originate in the gut. In this review, we discuss human studies that support ENS Lewy pathology as a characteristic feature of PD, and present evidence from humans and animal model systems that α-synuclein aggregation may follow a prion-like spreading cascade from enteric neurons, through the vagal nerve, and into the brain. Given the accessibility of the human gut to pharmacologic and dietary interventions, therapeutic strategies aimed at reducing pathological α-synuclein in the gastrointestinal tract hold significant promise for PD treatment.

## 1. Introduction

With a continually increasing disease burden of nearly 10 million patients worldwide [1], Parkinson’s disease (PD) is a devastating neurodegenerative disorder that is characterized by the loss of multiple neuronal populations and the aggregation of α-synuclein protein into intracellular inclusions known as Lewy bodies (LBs) [2,3]. The progressive death of dopaminergic neurons in the substantia nigra leads to a depletion of dopamine signaling in the striatum that manifests as classic Parkinsonian symptoms, such as bradykinesia, rigidity, and resting tremors which worsen over time [4]. In addition to impaired movement, PD patients also often experience non-motor symptoms, including depression, hyposmia, difficulty sleeping [5], dementia [6], and gastrointestinal issues [7]. Autonomic dysfunction can appear decades before the onset of motor signs [5,7], highlighting the complexity of the disorder and offering potential early points of intervention. While there is currently no cure for PD, administration of medications such as the dopamine precursor, levodopa, or surgical therapies such as deep brain stimulation can provide symptomatic relief, along with available treatments for non-motor indications.

The high occurrence of gastrointestinal dysfunction in the early stages of PD [7], coupled with increased recognition of PD gut microbiome dysbiosis [8] and repeated observations of α-synuclein pathology in the enteric nervous system (ENS) of PD patients [9,10,11,12,13,14,15,16,17,18,19,20,21,22,23], have collectively inspired great interest in the possibility that the disorder may originate in the gut. The gut-brain axis is a complex bidirectional signaling network by which the brain communicates with the gastrointestinal tract via the ENS that relays both sensory and motor information, bacteria-derived neuroactive molecules, and microbiome-induced cytokine release [8]. Disruption of the gut-brain axis can lead to a range of disorders from irritable bowel syndrome to functional gastrointestinal disorders, as well as potentially mood disorders and chronic pain [24]. In PD, patients experience a multitude of clinical symptoms that span the entirety of the gastrointestinal tract, including drooling, swallowing difficulties, delayed gastric emptying, small intestinal bacterial overgrowth, and constipation [7]. In addition, PD gut microbiota display an enrichment of species in the *Christensenella* [25], *Akkermansia* [26], and *Lactobacillus* [25,27] genera; depletion of species in *Bacteroides* [25,27], *Clostridium* [26,27], and *Faecalibacterium* [28,29] genera; and decreased levels of short-chain fatty acids [28]. Given that PD is a neurological disorder, these findings are consistent with gut-brain axis disturbances that may play a role in PD pathogenesis.

α-Synuclein aggregation is a hallmark of PD, yet the relationship of protein aggregation and neurodegeneration is still unclear despite extensive research efforts. Rare mutations in the α-synuclein gene, *SNCA*, cause early-onset forms of familial PD [30,31,32,33,34,35], and in sporadic PD (which accounts for at least 85% of cases), wild-type α-synuclein protein accumulates into LB inclusions and Lewy neurite (LN) axonal deposits [3,36]. In 2003, Braak et al. [37] hypothesized that PD could be staged by a topographical progression of α-synuclein lesions, the first of which appear in the dorsal motor nucleus (DMN) of the vagal nerve in the brainstem of pre-Parkinsonian patients. α-Synuclein pathology then spreads until it reaches the substantia nigra, coinciding with motor symptoms, and ultimately invades the neocortex, when patients may present with cognitive decline [36,37]. This progressive buildup of pathology across interconnected brain regions is consistent with prion-like transmission of α-synuclein from cell–cell, for which overwhelming evidence now exists in animal and cell culture model systems [38,39,40,41,42,43,44,45,46,47,48,49,50]. Furthermore, the early involvement of the DMN of the vagus, which serves as a major connection between the brain and the periphery, suggests that α-synuclein aggregates may first form in the ENS and gain access to the central nervous system (CNS) via the vagal nerve [14]. In this review, we will critically evaluate the evidence that ENS pathology may play a causative role in PD, with a focus on human studies and mammalian animal models of prion-like α-synuclein transmission. We will also bring to light the current gaps in knowledge and present new tools, including gut-to-brain *C. elegans* models, for advancing the scientific understanding of PD.

## 2. The Human Enteric Nervous System

The ENS is an intricate network of neuronal cell bodies and fibers that perform a wide array of digestive functions, including moving food through the gastrointestinal tract, facilitating nutrient uptake, regulating local blood flow, and supporting the immune system [51]. While the ENS is able to function independently of the CNS, bidirectional communication between the ENS and CNS serves to relay important information that ultimately affects organismal behavior and gastrointestinal functioning. The gut-brain axis involves the delivery of sensory information to the brain via spinal and vagal afferent pathways, and efferent motor signals to the gut by way of sympathetic and parasympathetic divisions of the autonomic nervous system [52].

Parasympathetic innervation via the vagal nerve originates in preganglionic neurons of the DMN, which synapse onto postganglionic neurons of the ENS. The nucleus ambiguus in the brainstem also supplies vagal motor efferents specifically to the pharynx and esophagus [53]. Vagal innervation of the ENS is densest in the upper gastrointestinal tract at the level of the esophagus and stomach and decreases more distally. With little to no vagal input to the distal colon and rectum, these regions are primarily regulated by the sacral parasympathetic nucleus of the spinal cord. The activity of the DMN and nucleus ambiguus can be modulated by sensory information that is relayed from the ENS through vagal afferent pathways to the nucleus of the solitary tract in the brainstem [53]. While the influence of parasympathetic pathways can result in both enhancement and suppression of gut motility, sympathetic innervation of the digestive tract mainly acts to inhibit motility as a pro-survival reflex that is mediated by prevertebral sympathetic ganglia [54].

Within the ENS, two main neuronal networks perform the complex integration of all local (intrinsic) neuronal activity, input from extrinsic sympathetic and parasympathetic neurons, as well as cues from the gastrointestinal environment. These networks, known as the myenteric (or Auerbach’s) plexus, and the submucosal (or Meissner’s) plexus, use the integrated input to determine their own sensory, motor, and secretory output. The ENS is located in the gastrointestinal wall, which is comprised of four main layers: the mucosa, submucosa, muscular layer, and adventitia or serosa (Figure 1). The layer closest to the lumen of the gut, the mucosa, can be further divided into the epithelium, which forms the lining of the mucosa; the lamina propria, which is made up of connective tissue; and the muscularis mucosae, a layer of smooth muscle. Adjacent to the mucosa, the submucosa contains connective tissue, lymphatic and blood vessels, and Meissner’s plexus, which exists predominantly in the walls of the small intestine and the colon. The next layer, the muscular layer, consists of circular and longitudinal smooth muscle sublayers, with Auerbach’s plexus situated in between the sublayers and present along the entire length of the gastrointestinal tract. The final layer of the gastrointestinal wall is mainly connective tissue, either adventitia or serosa, depending on whether there is attachment to the surrounding organs.

Within Auerbach’s and Meissner’s plexuses, there are a multitude of neuronal subtypes (Figure 1). In Auerbach’s plexus of the muscular layer, local motor circuits function to control gastrointestinal tract motility. These circuits are comprised of excitatory and inhibitory motor neurons that cause contraction and relaxation, respectively, of both the circular and longitudinal muscles. The excitatory motor neurons primarily use acetylcholine as their neurotransmitter, but also use tachykinins and other signaling molecules, while the inhibitory motor neurons use vasoactive intestinal polypeptide (VIP) and nitric oxide in addition to other neurotransmitters [55]. The excitatory and inhibitory motor neurons receive local input from myenteric interneurons and sensory information from intrinsic primary afferent neurons (IPANs). They are also subject to extrinsic modulation by cholinergic parasympathetic efferents arising from the preganglionic neurons of the DMN of the vagus. These vagal fibers are organized into parallel pathways that innervate either excitatory postganglionic motor neurons or inhibitory postganglionic motor neurons in the ENS [53].

In addition to this circuitry, VIP- or acetylcholine-producing secretomotor and vasodilator neurons can be found primarily in Meissner’s plexus, with innervation from IPANs, interneurons, and extrinsic signals from vagal efferent pathways. Unlike the parallel excitatory and inhibitory pathways from the DMN of the vagus to the motor circuits of the ENS, vagal innervation of secretory ENS neurons is primarily excitatory [53]. Secretomotor neurons control gastrointestinal secretions, while vasodilator neurons synapse onto local arterioles and regulate blood flow. Other ENS neuron types include intestinofugal neurons that synapse onto sympathetic ganglia, and motor neurons that innervate the muscularis mucosae [55].

## 3. Lewy Pathology in the Enteric Nervous System in PD

The earliest report of LB pathology in the ENS of PD patients was published in 1984 by Qualman et al. [9], who documented LBs in Auerbach’s plexus of the colon from one PD patient and the esophagus from another PD patient. The esophageal LBs were associated with ganglion cell degeneration. In 1987, Kupsky et al. [10] found LBs in the ganglion cells of both Auerbach’s and Meissner’s plexuses of the colon and rectum in a PD patient with megacolon. Following this work, a series of studies by Wakabayashi and colleagues found that LB pathology in PD is widely distributed throughout the ENS from the upper esophagus to the rectum in both Auerbach’s and Meissner’s plexuses [11,12,13]. The greatest LB burden was found in the Auerbach’s plexus of the lower esophagus [11,12,13], and LBs were primarily found in VIP-producing neurons although there was also rare colocalization of pathology with tyrosine hydroxylase-positive processes [12], which may correspond to noradrenergic sympathetic fibers. No loss of enteric neurons was noted [11,12].

A few years later in 1997, it was discovered that a principal component of LBs is the presynaptic protein, α-synuclein [3], and immunohistochemistry against α-synuclein or its modified forms quickly became the gold-standard for the detection of Lewy pathology. Braak and colleagues [37] used α-synuclein staining to define six stages of PD based on the stereotypical distribution of Lewy pathology in PD patient brains. This staging scheme postulated that, in early stages of disease, focal pathology is present in a small number of circumscribed brain regions, but as the disease progresses, pathology becomes more widespread, with the recruitment of additional areas in each advancing stage. In the first stage, Lewy pathology is present in the DMN of the vagal nerve, anterior olfactory nucleus, and olfactory bulb. In Stage 2, areas such as the raphe nuclei and locus coeruleus become affected, and in Stage 3, the amygdala, basal forebrain, and substantia nigra pars compacta begin to show α-synuclein lesions. The cerebral cortex becomes involved in Stage 4, with the mesocortex affected first and the neocortex showing pathology in Stages 5 and 6 [37].

Since the brain regions affected in the Braak PD staging scheme are synaptically interconnected, and one of the earliest afflicted regions is the DMN of the vagus connecting the brain to the ENS, Braak and colleagues further investigated α-synuclein in the ENS of autopsy cases staged for Lewy pathology in the CNS [14]. In total, ten cases were examined, three of whom had been diagnosed with sporadic PD and harbored Stage 4 or 5 brain pathology, two of whom were considered incidental or presymptomatic cases having Stage 2 or 3 PD pathology, and the rest having no evidence of Lewy pathology. α-Synuclein inclusions were found in the gastric wall and the DMN of all confirmed PD and incidental cases but none of the controls, and lesions were observed in both Auerbach’s and Meissner’s plexuses among the PD cases [14]. The presence of Lewy pathology in the ENS of incidental cases, which harbored early-stage PD brain pathology, is consistent with ENS involvement early in PD. Yet, the small number of cases examined precludes the ability to make conclusions regarding sequential involvement of the ENS and CNS.

Following these foundational studies, several investigations were undertaken to determine the extent to which ENS pathology is a characteristic of PD. Beach et al. [16] examined phosphorylation of α-synuclein (a disease-associated modification) in the peripheral nervous system of patients with LB disorders and controls. A total of 11/17 PD patients (65%) had α-synuclein pathology in the gastrointestinal tract compared with 0/23 controls. Consistent with earlier findings of LB distribution in PD [11,12,13], phospho-α-synuclein staining was more concentrated in the esophagus than in lower gastrointestinal regions, and Auerbach’s plexus appeared to be affected to a greater extent than Meissner’s plexus [16]. In a follow-up study [18], Lewy pathology was identified in the ENS of 13/13 PD patients and none of the controls, although the number of overlapping patients between the two studies is unclear [16,18]. LBs were not found in nitric oxide or VIP neurons, except for a single VIPergic neuron in Auerbach’s plexus that contained a LB [18]. There was low frequency (~3%) colocalization with tyrosine hydroxylase-positive fibers, but otherwise the cell types harboring LBs were not identified. In addition, neurodegeneration in Auerbach’s plexus was carefully examined along the length of the gastrointestinal tract in PD and controls, and the authors found no neuronal loss in PD [18]. This is in contrast to a different group which found that in 15/20 PD patients, but not in controls, intestinal ganglion cells showed atrophy and/or pycnotic nuclei, and all patients with ENS degeneration also had intestinal α-synuclein deposits [22]. The question of whether Lewy pathology in the ENS is associated with local neurodegeneration thus remains unsettled.

To date, the largest postmortem study of the ENS in PD patients and controls was conducted using the Brain Bank for Aging Research (BBAR) in Japan [23]. The lower esophagus was examined from 46 PD patients, which included 38 who were diagnosed with PD dementia and the related disorder, Dementia with Lewy Bodies. A total of 340 controls who did not have Parkinsonism, dementia, or evidence of degeneration in the locus coeruleus or substantia nigra were used for comparison. Remarkably, LBs and LNs were observed in 41/46 PD subjects (89%) compared with 0% of the controls. Since the upper alimentary canal was previously found to have greater LB burden in PD than lower regions [11,12,13,16], the BBAR study of the esophagus can be regarded as a highly relevant sampling of tissue. Across all postmortem studies of the ENS reviewed here, PD patients had more Lewy pathology in the esophagus, stomach, small intestine, large intestine, and rectum, than controls (Figure 2A), and the total number of PD patients harboring LBs in any segment of the gastrointestinal tract (165/212; 78%) far exceeds that in controls (17/618; 3%) (Figure 2B). These findings, therefore, support the notion that α-synuclein aggregation in the ENS is a characteristic feature of PD.

## 4. Does α-Synuclein Aggregation Begin in the Gut and Spread to the Brain?

Despite ample evidence of ENS pathology in PD and the hypothesis that it may precede affliction of the CNS [14], it remains unknown if α-synuclein aggregation follows a prion-like spreading cascade from enteric neurons, through the vagal nerve, and into the brain. α-Synuclein exhibits many prion-like properties, including the ability of aggregated conformations to self-replicate by inducing, or ‘seeding’, the aggregation of physiological α-synuclein [57,58]. In this way, the aggregate load within a cell can become amplified. In addition, similar to prions, α-synuclein can escape from one cell and infect a neighboring cell, as has been shown in culture systems and rodent transplantation studies in which host α-synuclein was detected in grafted cells [47,49]. Strikingly, postmortem studies of PD patients who received fetal mesencephalic grafts into the striatum showed that 11–16 years following transplantation, grafted neurons contained LB-like α-synuclein inclusions [59,60]. Given the young age of the grafted cells, it is unlikely that pathology developed independently of exposure to α-synuclein and/or other factors from the diseased host tissue, thereby suggesting that cell–cell transmission of α-synuclein in humans is possible.

### 4.1. Evidence from Incidental LB Disease and Prodromal PD

Incidental LB disease (ILBD) is thought to be a precursor to the development of PD, with subclinical LB pathology in the DMN of the vagus or other brain regions consistent with early Braak PD stages. If ILBD is in fact an incipient form of PD, then evaluation of the ENS for α-synuclein inclusions may shed light on whether protein aggregation in PD is of central or peripheral origin. Indeed, this was the motivation for Bloch et al. [15], who examined 17 cases of ILBD postmortem and found 14 of them (82%) had α-synuclein pathology in the ENS of the esophagus. Lesions were found mainly in Auerbach’s plexus but also in Meissner’s plexus [15]. In a larger study, esophageal Lewy pathology was found in 36/103 (35%) of preclinical or prodromal LB disease subjects on autopsy compared with 0/340 controls [23]. Several additional postmortem reports have documented Lewy pathology in the ENS of ILBD cases, albeit with very small sample sizes [9,13,14,16,17,18], including one study that noted LBs or LNs in the stomach, small intestine, large intestine, and rectum from at least one ILBD case for each gastrointestinal segment [18]. In general, ILBD cases have tended to have less severe LB pathology than diagnosed PD cases [16,18].

While these findings collectively suggest that α-synuclein aggregates may be present in the ENS at early PD stages, consistent with a peripheral initiation of pathology, it is not possible to confirm a specific sequential progression of α-synuclein disease from postmortem (endpoint-only) studies. Stokholm et al. [61], therefore, examined tissue blocks collected during routine gastrointestinal biopsies and surgical resections from patients who were later diagnosed with PD (on average, 7 years from diagnosis). Phosphorylated α-synuclein was observed in 22/39 (56%) of these prodromal PD patients, which was significantly higher than controls (23/90, 26%). Strikingly, α-synuclein pathology was detected in neuronal structures of the gastrointestinal tract up to 20 years prior to PD diagnosis, suggesting that ENS involvement might precede that in the CNS by decades [61]. Importantly, however, concurrent analysis of ENS and brain pathology is not possible in living patients, and, therefore, it is still not known if inclusions exist in the human ENS prior to their appearance in the CNS.

### 4.2. Evidence from Human Vagotomy Studies

Besides the presence of Lewy pathology in the ENS, another key aspect of the gut-to-brain hypothesis of PD is the transmission of α-synuclein through the vagal nerve. The higher density of ENS pathology in the esophagus and stomach compared to lower regions in PD [11,12,13,16] is consistent with the high concentration of vagal inputs to these areas. PD patient vagotomy studies may, therefore, shed light on the dependence of the disease on intact vagal innervation of the gut. A former treatment for peptic ulcer, vagotomy, is the severing of the vagal nerve either fully (as in truncal vagotomy), solely to the stomach (selective vagotomy), or most selectively to the fundus and body of the stomach only (superselective vagotomy). In 2015, Svensson et al. [62] analyzed the risk of developing PD in over 11,000 patients catalogued in the Danish National Patient Registry as having undergone a vagotomy procedure. Truncal and selective vagotomy patients were grouped as one cohort (*n* = 5339), but despite this potential limitation, this group was found to have reduced risk of developing PD compared with over 60,000 controls. Notably, the effect was strongest for subjects with over 20 years of follow-up from vagotomy [62], consistent with a long latency of prodromal disease. Superselective vagotomy, which spares some connections between the stomach and the brain, as well as intestinal vagal innervation, was not associated with protection from PD [62], potentially due to the remaining vagal routes available for α-synuclein transmission.

Although these findings were challenged by a group that re-analyzed data from the same source [63], the re-examination had significant flaws. Importantly, truncal and selective vagotomies were regarded as two separate groups despite a known lack of reliability for these classifications in the Danish registry [62]. In addition, there was no minimum cut-off time from vagotomy to PD diagnosis, thereby potentially including prodromal PD patients who received vagotomy too late for their disease to be modified [64]. Moreover, a different group analyzing over 9,000 vagotomy patients in a Swedish cohort also found that truncal vagotomy reduced the risk of PD [65]. Thus, while only a few studies have examined the potential relationship of vagal denervation with PD, the findings generally support a protective effect, potentially by reducing the ability of α-synuclein to invade the CNS.

### 4.3. Prion-Like Transmission of α-Synuclein in Rodents and Monkeys

There is now extensive evidence from animal models supporting the theory of prion-like transmission of α-synuclein in PD and other synucleinopathies. In mice and rats, intracerebral inoculation of recombinant α-synuclein pre-formed fibrils (PFFs), brain tissue from symptomatic α-synuclein transgenic mice, or brain tissue from human synucleinopathy patients, results in widespread deposition of LB-like inclusions that are often associated with neurodegeneration and motor dysfunction [38,40,41,42,45,46,47,50]. Moreover, it appears that, regardless of the site of injection, aggregation propagates along synaptic connections and requires the presence of endogenous α-synuclein, similar to prions [38,40,42,45,46,50]. In macaque monkeys, injection of PD brain tissue containing insoluble LBs into the substantia nigra or striatum caused a loss of striatal terminals followed by dopamine neuron death and diffuse α-synuclein deposits in the remaining nigral cells [40]. Other studies have also shown that α-synuclein pathology can spread to the CNS following intravenous [38], intramuscular [39], or intraperitoneal [66] injection of recombinant α-synuclein aggregates into rodents.

Importantly, the ability of α-synuclein to propagate from the gut to the brain has also been demonstrated in rodent models. In one study, Holmqvist et al. [67] injected PD substantia nigra lysate or recombinant human α-synuclein fibrils into the intestinal wall of adult rats. In both cases, at 12 h post-injection, human α-synuclein was detected in the intestinal wall but not in the vagal nerve. By 48 and 72 h post-injection, however, human α-synuclein immunoreactivity could be seen in the vagal nerve of animals injected with PD lysate or recombinant fibrils, but not in controls that were injected with bovine serum albumin. Human α-synuclein was also detected in the DMN of the vagus following intestinal injection of either PD brain lysate, or monomeric, oligomeric, or fibrillar forms of recombinant α-synuclein [67]. These data strongly support the ability of α-synuclein of gastrointestinal origin to spread via the vagal nerve to the DMN in the brainstem. Similarly compelling evidence for this path of gut-to-brain α-synuclein spread was reported by Kim et al. [68], who injected mouse α-synuclein PFFs into the pylorus region of the stomach and duodenum of the small intestine in wild-type mice. Following gastrointestinal PFF injection, phosphorylated α-synuclein in the CNS was first detected in the DMN of the vagus and then in the locus coeruleus, amygdala, substantia nigra, and eventually the prefrontal cortex, closely mirroring the Braak staging scheme for PD. The accumulation of α-synuclein pathology was also associated with a progressive loss of dopaminergic neurons, motor dysfunction, and cognitive symptoms. Remarkably, animals that underwent truncal vagotomy or were lacking endogenous α-synuclein were entirely protected from the spread of α-synuclein pathology and any of its associated toxicities [68]. These findings further support the notion that gut-derived α-synuclein is capable of propagating through the vagal nerve in a prion-like manner to induce CNS disease.

Additional studies provide further evidence in favor of this hypothesis. Challis et al. [69] found that PFF inoculation into the duodenum of aged mice causes gastrointestinal dysfunction and promotes α-synuclein pathology in the brainstem, coupled with motor decline and decreased striatal dopamine. In two reports from Uemura et al. [70,71], injection of PFFs into the gastric wall of either wild-type [70] or α-synuclein transgenic mice expressing the A53T familial PD mutation [71] resulted in phosphorylated α-synuclein lesions in the DMN of the vagus. Similar to Kim et al. [68], wild-type animals who received hemivagotomy prior to PFF injection had α-synuclein deposition only on the unvagotomized side, consistent with a vagal-dependent spread of pathology [70]. There were two groups that have also shown that oral delivery of α-synuclein fibrils can induce neurological symptoms and Lewy-like pathology in the CNS of heterozygous A53T α-synuclein transgenic mice [72,73]. Specifically, α-synuclein pathology was observed in the spinal cord, brainstem, and substantia nigra following oral administration of aggregated α-synuclein [72,73]. In addition, A53T transgenic mice were also found to harbor seeding-competent α-synuclein in colon tissue several months before similar species were detected in the brain, suggesting that pathological α-synuclein can form in the ENS prior to appearing in the CNS [74].

In some cases, gut-initiated α-synuclein spreading to the CNS was reported to be a transient phenomenon. Manfredsson et al. [75] injected rats and non-human primates with PFFs into the descending colon, and in both cases, observed abundant ENS pathology that persisted even after 1 year. In rats, minor α-synuclein pathology was also present in the DMN of the vagus and the locus coeruleus at 1-month post-injection, but this was not observed at later timepoints, and CNS lesions were not observed at any timepoint in the non-human primates. Since only ~20% of ENS ganglia in the descending colon receive vagal innervation [75], it is perhaps not surprising that little or no spread of pathology was observed in the brainstem of either the rats or monkeys. Moreover, the presence and then disappearance of the CNS lesions in this and other studies [70,71] argues that a certain threshold of α-synuclein transmission must be met in order to have sustained CNS pathology. Beneath this threshold, it is likely that cellular degradation mechanisms are able to clear the misfolded α-synuclein before further aggregation and spreading can occur.

### 4.4. C. elegans as a Powerful Model System to Study α-Synuclein Pathogenicity in PD

While rodent and non-human primate models provide essential information with regards to how α-synuclein can behave in a mammalian system, complementary animal models that offer a rapidly aging nervous system and high genetic tractability are necessary to accelerate the discovery of disease mechanisms and potential treatments. The small nematode worm, *C. elegans,* provides such a platform, having a well-defined nervous system that gives rise to a complex set of behaviors [76], orthologs for 60–80% of human genes [77], conserved neurotransmitter signaling [78], and suitability to rapid large-scale behavioral and phenotypic screening approaches [76]. *C. elegans* is a premier model system to study aging and age-related disease, due to its short lifespan (2-4 weeks) and stereotyped age-dependent decline at the tissue, cellular, and molecular levels [79]. In addition, transgenic expression of human α-synuclein in worms has recapitulated progressive age-dependent neuron death, protein aggregation, and behavioral deficits [80,81,82], and proven useful for the study of cell autonomous disease mechanisms in dopaminergic neurons [83,84,85].

Several studies have also demonstrated cell–cell transmission of human α-synuclein in *C. elegans* [86,87,88]. Bimolecular fluorescence complementation (BiFC) is one technique that has been used to visualize α-synuclein transfer between cells. In this approach, α-synuclein in one group of cells is fused to the N terminus of a fluorophore, while α-synuclein in another group of cells is fused to the C terminus of the fluorophore. Thus, fluorescence should only occur if α-synuclein fusion proteins are able to translocate from one cell to another and come in sufficient proximity for the fluorophore to assemble. Using BiFC in *C. elegans*, α-synuclein has been shown to transfer from neuron–neuron in synaptically connected circuitry [86], and to travel bidirectionally between neurons and pharyngeal muscle cells [87]. The latter observation was associated with enhanced neurodegeneration, functional decline, and decreased lifespan, suggesting that α-synuclein transmission is toxic [87]. In another study, α-synuclein was observed to translocate from donor dopaminergic neurons or muscle cells to recipient hypodermis tissue [88]. While these findings indicate that cell–cell α-synuclein transmission can occur in *C. elegans*, the aggregation state of α-synuclein was often not evaluated, and critically, there remains a need for specifically gut-to-brain PD worm models.

In an effort to generate prion-like α-synuclein transmission models initiated in the gut of *C. elegans*, our group recently published the neurotoxic effects of feeding worms human α-synuclein PFFs [89]. To our knowledge, this is the first report of α-synuclein PFF exposure in *C. elegans*. Similar to mouse models, we found that PFF ingestion in *C. elegans* promotes dopaminergic neurodegeneration, accelerates the aggregation of host α-synuclein in muscle, and induces an age-dependent motor decline. Importantly, monomeric α-synuclein feeding was unable to produce the same effects, and PFF-fed worms lacking host α-synuclein were also protected [89]. These findings are consistent with prion-like propagation of disease by gut-derived α-synuclein fibrils, potentially via the worm alimentary nervous system, which remains to be tested. Our results are also consistent with previous reports of diet-induced α-synuclein aggregation in *C. elegans* [90]. Specifically, worms fed *E. coli* that produce the bacterial amyloid, curli, showed increased aggregation of α-synuclein in muscle cells. In the same study, aged rats that were fed curli-producing bacteria showed an increase in α-synuclein deposition in the ENS, as well as in the hippocampus and striatum [90].

To identify potential regulators of gut-to-brain synucleinopathy, our group used the rapid and highly manipulable *C. elegans* PFF models to conduct a targeted RNAi screen [89]. Previously, heparan sulfate proteoglycans (HSPGs) were identified as potential cell surface receptors for the internalization of α-synuclein PFFs in vitro [91]. We tested seven genes in the highly conserved HSPG pathway and showed that of those, five genes, namely *SDC1/sdn-1* (cell surface membrane-bound proteoglycan syndecan), *EXT1/rib-1* (exostosin glycosyltransferase 1), *EXTL3/rib-2* (exostosin like glycosyltransferase 3), *NDST1/hst-1* (N-deacetylase and N-sulfotransferase 1), and *HS3ST6/hst-3.2* (heparan sulfate-glucosamine 3-sulfotransferase 6), were necessary for PFF-dependent motor dysfunction [89]. *SDC1/sdn-1, EXT1/rib-1,* and *NDST1/hst-1* were also required for PFF-induced host α-synuclein aggregation and dopamine neuron degeneration [89]. These results constitute the first in vivo evidence that HSPGs can regulate disease phenotypes caused by α-synuclein PFFs and suggest that the *C. elegans* PFF models can be used in future studies to further uncover the mechanisms of α-synuclein spread from the gut and resulting neurotoxicity.

### 4.5. Alternative Hypotheses of α-Synuclein Spreading in PD

Despite mounting evidence in humans and animal models supporting the gut-to-brain hypothesis of α-synuclein transmission in PD, alternative possibilities have been proposed that fuel ongoing debate. A major criticism of the gut-origin hypothesis of PD is the lack of individuals found to have α-synuclein pathology in the ENS in the absence of pathology in the CNS. It would be expected that if α-synuclein pathology begins in the ENS and spreads to the CNS via the vagal nerve, there should be normal subjects with undiagnosed, prodromal PD that harbor ENS and/or vagal nerve pathology without evidence of lesions in the CNS. To address this issue, Beach and colleagues [56] conducted an autopsy study of stomach and/or vagal nerve tissue from 111 normal elderly controls that had no CNS pathology, 33 ILBD cases with some CNS pathology, and 53 confirmed PD cases. None of the normal subjects were found to have α-synuclein lesions in the stomach or vagal tissue, whereas 17% and 81% of ILBD and PD cases, respectively, had stomach pathology, and 46% and 89% of ILBD and PD cases, respectively, had vagal pathology. The authors concluded that, once again, the lack of α-synuclein inclusions in the ENS/vagal nerve of normal control subjects argues against a gut-first hypothesis of PD, and instead supports a CNS origin of disease [56].

Several limitations of the study weaken this conclusion, however. If the gut-to-brain hypothesis is true, then the identification of ENS/vagal-positive α-synuclein cases among normal controls in a given study is predicated on the existence of prodromal PD cases within this group. The rate of PD among the aged (65 and older) population in the US is estimated to be 1.6% [92]. If this rate is also representative of prodromal PD in the US, then a sample size of 111 controls, as in the Beach et al. [56] study, would be expected to include at most only 2 subjects with ENS/vagal pathology (and no CNS pathology). Even with an estimated 90,000 new PD cases each year among 65 and older individuals in the US [93], and assuming a 20-year prodromal period [61], 1.8 million potential prodromal cases represent 3.2% of the 56 million people aged 65 and over in the US [94], corresponding to only 4 prodromal cases being included at most in the control group of the Beach study. Therefore, the low rate of prodromal PD estimated to exist in the aged population drastically reduces the probability of detection. Moreover, the Beach et al. [56] study only examined stomach tissue and did not look at other gastrointestinal organs, and similarly only sampled one area of the vagus nerve. These limitations notwithstanding, Beach and coauthors investigated an important question in a large human cohort, albeit probably not large enough to make firm conclusions regarding prodromal PD.

Rather than arguing specifically against the gut-to-brain hypothesis of PD, the discovery of pathological α-synuclein in the vagus nerve of the majority of PD patients and almost half of ILBD patients [56] can alternatively be interpreted as supporting the vagus nerve acting as a conduit for α-synuclein transmission between the gut and the brain, potentially in either direction. Consistent with this, Borghammer and colleagues have put forth the hypothesis that PD cases fall into two main categories based on the trajectory of α-synuclein spread: brain-first versus body-first [95]. In brain-first PD, it is postulated that α-synuclein aggregation begins in the olfactory bulb or amygdala and eventually spreads to the ENS via anterograde transmission through the vagal nerve. In body-first PD, pathological α-synuclein arises first in the ENS and spreads to the CNS via retrograde transmission through the vagal nerve. Clinical evidence supports the grouping of PD patients into these categories [95], and animal studies have shown that α-synuclein is capable of travelling bidirectionally through the vagal nerve [96]. It is, therefore, likely that multiple subtypes of PD exist, making the probability of identifying prodromal body-first PD in normal control subjects even lower than predicted above.

## 5. Conclusions

Ample evidence now suggests that α-synuclein can act similarly to prions and can potentially infiltrate the CNS from peripheral tissues such as the gastrointestinal tract. It is likely that multiple sites of initiation exist for the primary α-synuclein misfolding event to occur, with subsequent amplification of pathological aggregates, escape into the extracellular space, and infection of neighboring cells within synaptically connected networks. The vagal nerve offers a direct route by which α-synuclein may be able to propagate from the ENS to central brain regions, and both human and animal studies support this pathway as a prime candidate for PD progression.

Critical questions remain to be answered, however. At the level of the ENS, it is still unknown which cell types contain α-synuclein inclusions and could theoretically transmit pathological species through interconnected circuitry. Although VIPergic neurons were originally identified as a major cell type harboring LBs in PD [12], this finding was not reproduced in a later study [18]. Conflicting data also exist regarding potential enteric neuron degeneration in PD. At least two studies have found enteric ganglion cell degeneration [9,22], while several others have reported no evidence of neuronal cell loss in the ENS [11,12,18]. In addition, a major outstanding question is what factors might initially trigger α-synuclein misfolding in the ENS. Braak et al. [97] originally proposed that an unknown pathogen in the gut, such as a virus or prion particle, could induce α-synuclein aggregation in enteric neurons. This kind of pathogen, potentially α-synuclein itself introduced from the diet [98], might be able to gain entry across the gut epithelial barrier under conditions of increased permeability, such as “leaky gut” caused by inflammation or infection [98].

Another critical question is whether ENS pathology can be used as a biomarker for the early detection of PD. The discovery of a reliable biomarker in pre-Parkinsonian patients would be highly clinically significant, allowing for the potential of an early intervention that could slow or even prevent further disease. Despite extensive research efforts, the utility of α-synuclein detection in gastrointestinal biopsies and surgical resections remains unclear [99]. Methodological issues have posed great challenges, including the insufficient sensitivity and specificity of some α-synuclein immunohistochemical approaches for distinguishing PD patients from controls [21]. Still, there are studies that show promising results, such as a recent report of duodenal biopsies examined using the conformation-specific 5G4 α-synuclein antibody that preferentially detects aggregates [100]. This study found greater α-synuclein aggregation in PD cases compared with controls and detected pathology in both early- and late-stage PD patient biopsies [100].

A depiction of the gut-to-brain hypothesis of PD highlighting these and other unresolved questions can be found in Figure 3. The development of new animal models, such as gut-to-brain PD models in *C. elegans* [89], and new technologies including gut- and brain-specific organoids [101] may serve to complement existing rodent model systems by acting as platforms for high-throughput discovery. In all, the theory of gut-to-brain transmission of α-synuclein in PD is supported by a compelling body of evidence and warrants further study to determine its precise clinical relevance.

## Figures and Tables

**Figure 1 ijms-24-07205-f001:**
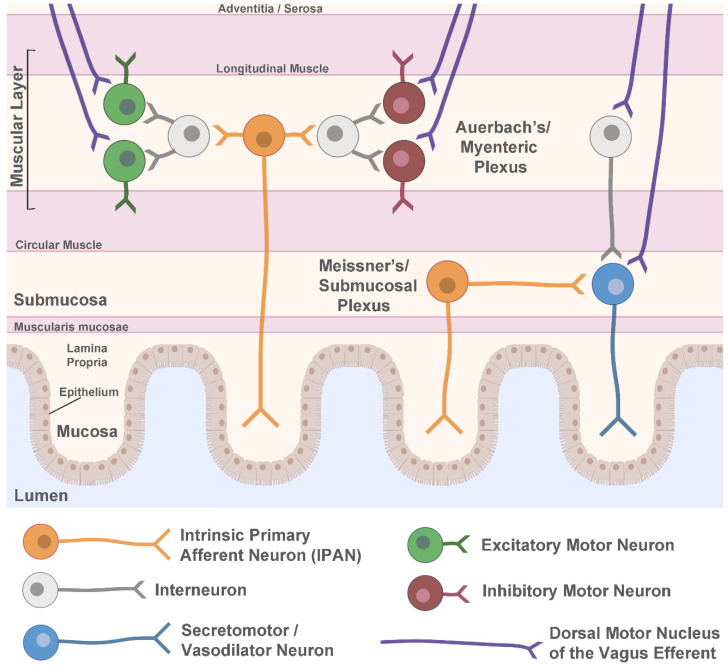
Schematic of the enteric nervous system. Shown are the major layers of the gastrointestinal wall with a simplified representation of neuronal circuitry. The myenteric (Auerbach’s) plexus contains motor circuits that control contraction and relaxation of the muscle layer whereas secretomotor and vasodilator neurons are primarily in the submucosal (Meissner’s) plexus and control local blood flow and secretion. Both plexuses receive extrinsic parasympathetic innervation from the dorsal motor nucleus of the vagus to help regulate gut motility. Sympathetic innervation and vagal sensory afferents are not shown. Figure created using Biorender.com.

**Figure 2 ijms-24-07205-f002:**
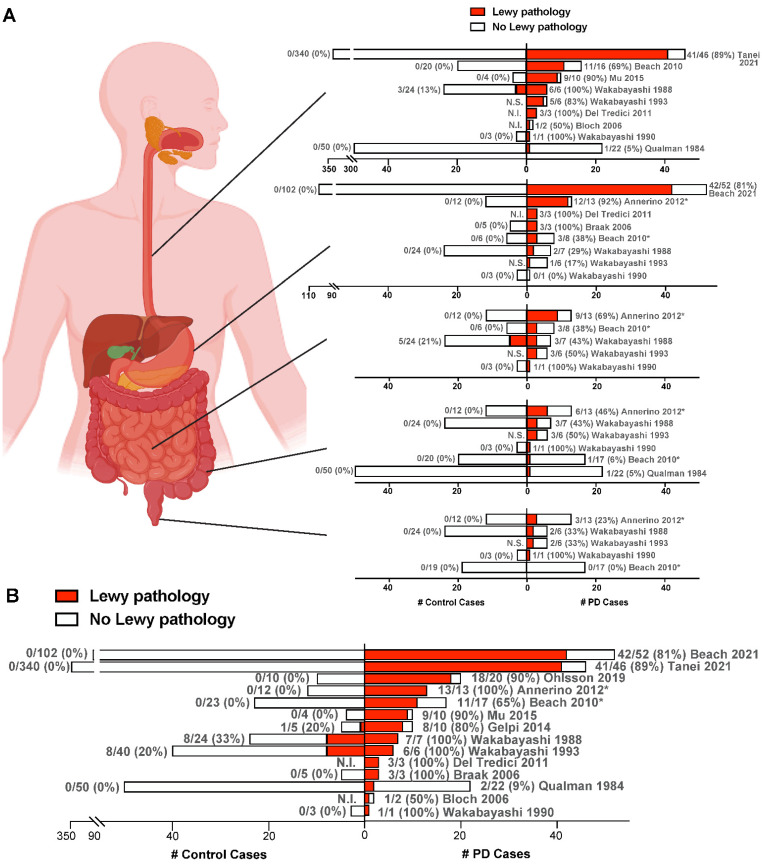
ENS Lewy pathology detected in postmortem studies of confirmed PD cases. (**A**,**B**) Each row represents one study, with the citation indicated. PD cases are to the right of the y-axis, while control cases are to the left of the y-axis. For each patient group, the number of cases with Lewy pathology (red) out of the total number of cases examined in that group is given next to the corresponding bar, with % positive for Lewy pathology given in parentheses. Only cases unique to each study are included. In (**A**), graphs (from top to bottom) represent studies of the esophagus, stomach, small intestine, large intestine, and rectum. In (**B**), for each study, the total number of PD or control cases with Lewy pathology detected in any gastrointestinal segment is shown. * Indicates an unknown number of subjects from these studies are overlapping. N.I., Not included in the study. N.S., Not shown in the study. Figure created using Biorender.com and GraphPad Prism 9. Refs. [9,11,12,13,14,15,16,17,18,19,20,22,23,56].

**Figure 3 ijms-24-07205-f003:**
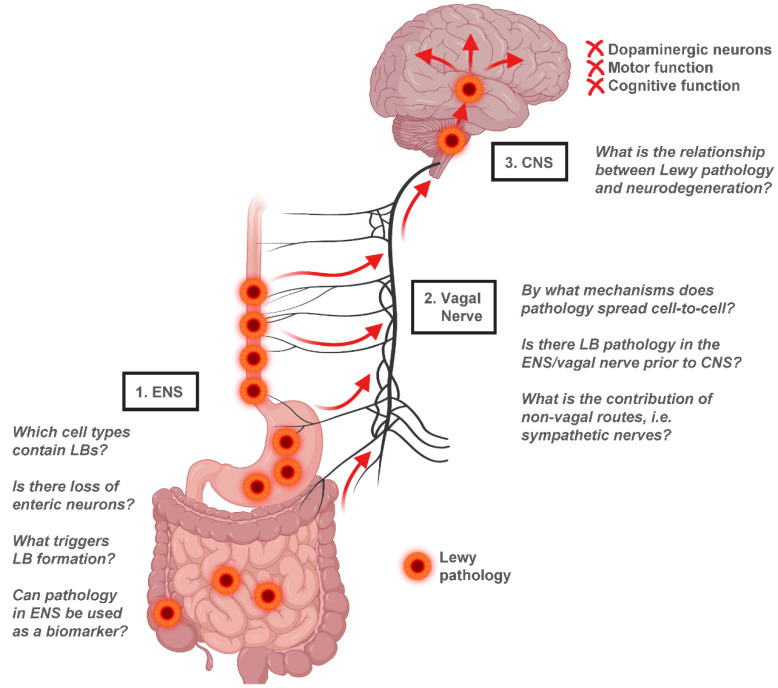
Gut-to-brain α-synuclein transmission hypothesis of PD. Evidence from human studies suggests that Lewy pathology in the ENS of PD patients is concentrated in the upper gastrointestinal tract, with the greatest LB density found in the esophagus and stomach. This distribution parallels vagal innervation of the ENS, consistent with a potential transmission of α-synuclein from the ENS retrogradely through the vagal nerve into the CNS. Once in the CNS, Lewy pathology appears to spread from the brainstem to the midbrain and finally to cortical regions, resulting in neurodegeneration and functional decline. Several unresolved questions relating to the gut-to-brain hypothesis of PD are highlighted. Figure created using Biorender.com.

## Data Availability

Not applicable.

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
