# Peer review of "Gut-to-Brain α-Synuclein Transmission in Parkinson’s Disease: Evidence for Prion-like Mechanisms"

_ijms, 2023, doi:10.3390/ijms24087205_

Round 1
Reviewer 1 Report
This review nicely summarised and updated current views in related to a critical topic in the research area of synucleinpathology. It is very well-written and easy to read. Would be even better if the authors could include a diagram or table to highlight the key messages. There is also a very recent publication that potentially might be interesting to include (PMID: 36847308).
Author Response
We thank the reviewer for their insightful feedback. Please see below for a point-by-point response to the reviewer's comments.
This review nicely summarised and updated current views in related to a critical topic in the research area of synucleinpathology. It is very well-written and easy to read.
We thank the reviewer for their positive comments.
Would be even better if the authors could include a diagram or table to highlight the key messages.
We agree, and we have now added an additional figure (Figure 3) that depicts the gut-to-brain hypothesis of PD.
There is also a very recent publication that potentially might be interesting to include (PMID: 36847308).
We have now included this reference in the Conclusions section.
Reviewer 2 Report
The work is very interesting and touches on important topics currently discussed in science. Especially nowadays, where neurodegenerative diseases affect more and more people. Parkinson's disease is one of the most common neurodegenerative diseases where, despite pharmacology, it is a progressive disease. Therefore, the search for not only potential drugs but also pathophysiology is important. The review is well prepared. It is worth enriching it with graphics explaining the potential relationships and pathomechanism of the disease
Author Response
We thank the reviewer for their insightful feedback. Please see below for a point-by-point response to the reviewer's comments.
The work is very interesting and touches on important topics currently discussed in science. Especially nowadays, where neurodegenerative diseases affect more and more people. Parkinson's disease is one of the most common neurodegenerative diseases where, despite pharmacology, it is a progressive disease. Therefore, the search for not only potential drugs but also pathophysiology is important. The review is well prepared.
We thank the reviewer for their positive comments.
It is worth enriching it with graphics explaining the potential relationships and pathomechanism of the disease
We agree, and we have now added an additional figure (Figure 3) that depicts the gut-to-brain hypothesis of PD.
Reviewer 3 Report
General comments
The Review by Merry Chen and Danielle E. Mor explores the evidence of ENS pathology in PD and discuss prion-like α-synuclein transmission studies in rodents and monkey models. The authors also highlight as C. elegans model can be useful to study α-synuclein pathogenicity in PD. The manuscript may be useful to take stock of current knowledge that supports a gut origin in PD and beneficial for IJMS readers. However, there are a few items that should be addressed:
Point 1: The concept that maybe PD starts in the gut is wide spread, but there are dissenting opinions. The review may be more complete if a brief chapter on olfactory bulb or brain origin and is included to comparing the current knowledge that support the brain-gut hypothesis.
Point 2: The authors mention in lines 234-237 that “Importantly, however, concurrent analysis of ENS and brain pathology is not possible in living patients, and therefore it is still not known if inclusions exist in the ENS prior to their appearance in the CNS”. They are right, however one recent study, which used A53T mutant mice, detected α-synuclein seeding activity in the ENS of pre-symptomatic mice prior to the appearance of seeding activity in the brain [Han, J.Y., C. Shin, and Y.P. Choi. Viruses, 2021. 13 (5). This finding should be discussed in the text by the authors.
Author Response
We thank the reviewer for their insightful feedback. Please see below for a point-by-point response to the reviewer's comments.
The Review by Merry Chen and Danielle E. Mor explores the evidence of ENS pathology in PD and discuss prion-like α-synuclein transmission studies in rodents and monkey models. The authors also highlight as C. elegans model can be useful to study α-synuclein pathogenicity in PD. The manuscript may be useful to take stock of current knowledge that supports a gut origin in PD and beneficial for IJMS readers.
We thank the reviewer for their positive comments.
However, there are a few items that should be addressed: Point 1: The concept that maybe PD starts in the gut is wide spread, but there are dissenting opinions. The review may be more complete if a brief chapter on olfactory bulb or brain origin and is included to comparing the current knowledge that support the brain-gut hypothesis.
We agree, and we have now added a section (section 4.5) discussing alternative hypotheses for alpha-synuclein spreading in PD, including olfactory/brain origin.
Point 2: The authors mention in lines 234-237 that “Importantly, however, concurrent analysis of ENS and brain pathology is not possible in living patients, and therefore it is still not known if inclusions exist in the ENS prior to their appearance in the CNS”. They are right, however one recent study, which used A53T mutant mice, detected α-synuclein seeding activity in the ENS of pre-symptomatic mice prior to the appearance of seeding activity in the brain [Han, J.Y., C. Shin, and Y.P. Choi. Viruses, 2021. 13 (5). This finding should be discussed in the text by the authors.
We thank the reviewer for this suggestion. We have now included a discussion of this study in section 4.3 about animal models.
Reviewer 4 Report
This is a very well-written review on the subject of the gut-brain axis and Parkinson's disease.
A brief addition on the different innervation of the different gut sections would be helpful in section 2 "The human enteric nervous system" to better understand the Lewis pathologies in these sections in relation to the gut-brain axis.
The summary of the current research is great, but the conclusion at the end is a bit weak in comparison.
Additions could be made here to include clinical relevance.
Overall, however, despite these suggestions, I definitely recommend the maunscript for publication.
Author Response
We thank the reviewer for their insightful feedback. Please see below for a point-by-point response to the reviewer's comments.
This is a very well-written review on the subject of the gut-brain axis and Parkinson's disease.
We thank the reviewer for their positive comments.
A brief addition on the different innervation of the different gut sections would be helpful in section 2 "The human enteric nervous system" to better understand the Lewis pathologies in these sections in relation to the gut-brain axis.
We thank the reviewer for this suggestion and have now added much greater detail of the innervation of the ENS to section 2, and more clearly stated throughout the manuscript how this innervation may relate to the distribution of Lewy pathology in PD.
The summary of the current research is great, but the conclusion at the end is a bit weak in comparison.
Additions could be made here to include clinical relevance.
We agree, and we have now added an additional figure (Figure 3) that is discussed in the Conclusion section, as well as a discussion of clinical relevance in terms of ENS pathology as a potential biomarker for PD.
Overall, however, despite these suggestions, I definitely recommend the maunscript for publication.
We thank the reviewer for their positive comments.